# Influence of Warming and Atmospheric Circulation Changes on
# Multidecadal European Flood Variability
Stefan Brönnimann,[1,2,*] Peter Stucki,[1,2] Jörg Franke,[1,2] Veronika Valler,[1,2] Yuri Brugnara,[1,2] Ralf
Hand,[1,2] Laura C. Slivinski,[3,4] Gilbert P. Compo,[3,4] Prashant D. Sardeshmukh,[3,4] Michel Lang,[5] Bettina
Schaefli[1,2]
[1] Oeschger Centre for Climate Change Research, University of Bern, Switzerland
[2] Institute of Geography, University of Bern, Switzerland
[3] University of Colorado, CIRES, Boulder, USA
[4] NOAA Physical Sciences Laboratory, Boulder, USA
[5] INRAE, Lyon-Villeurbanne, France
* corresponding author: stefan.broennimann@giub.unibe.ch
**Abstract**
European flood frequency and intensity change on a multidecadal scale. Floods were more frequent in
the 19[th] (Central Europe) and early 20[th] century (Western Europe) than during the mid-20[th] century and
again more frequent since the 1970s. The causes of this variability are not well understood and the
relation to climate change is unclear. Palaeoclimate studies from the northern Alps suggest that past
flood-rich periods coincided with cold periods. In contrast, some studies suggest that more floods
might occur in a future, warming world. Here we address the contribution of atmospheric circulation
and of warming to multidecadal flood variability. For this, we use long series of annual peak
streamflow, daily weather data, reanalyses, and reconstructions. We show that both changes in
atmospheric circulation and moisture content affected multidecadal changes of annual peak
streamflow in Central and Western Europe over the past two centuries. We find that during the 19[th]
and early 20[th] century, atmospheric circulation changes led to high peak values of moisture flux
convergence. The circulation was more conducive to strong and long-lasting precipitation events than
in the mid-20[th] century. These changes are also partly reflected in the seasonal mean circulation and
reproduced in atmospheric model simulations, pointing to a possible role of oceanic variability. For
the period after 1980, increasing moisture content in a warming atmosphere led to extremely high
moisture flux convergence. Thus, the main atmospheric driver of flood variability changed from
atmospheric circulation variability to water vapour increase.

**1. Introduction**

Changes in flood frequency and intensity depend on many factors (Hall, 2014; Tarasova, 2019) including changes in atmospheric processes such as moisture flux, convection, precipitation at different time scales, changes in hydrological processes such as infiltration or overland flow, the seasonal coincidence of snow melt and heavy precipitation, and on human interventions such as river bed and lake regulations, hydropower plants or other hydraulic constructions. Some of these factors are affected by climate change, but also multidecadal variations of climate play a role. During the 19th century, floods were more frequent in Alpine countries (Glaser et al., 2004, 2010; Brázdil et al., 2005; Blöschl et al., 2020, Schmocker-Fackel and Naef, 2010a,b; Himmelsbach et al., 2015; Lang et al., 2016) triggering political discussion that led to legislation on forest conservation and hydraulic engineering (Summermatter, 2005). In contrast, floods were comparably rare in Central Europe in the mid-20th century, a period when large infrastructure projects were planned and carried out (Pfister 2009). The causes of this multidecadal flood variability are not well understood. Atmospheric circulation changes played a role (Jacobeit et al., 2003; Mudelsee et al., 2004; Quinn and Wilby, 2013; Brönnimann et al., 2019), but this has not been well quantified. Furthermore, the relation to climate change is unclear. In this paper we analyse multidecadal flood variability in Europe in relation to atmospheric processes and in particular their link to climate change.

Better understanding this relation is relevant for assessing future flood risk. In that context, it is important to note that palaeoclimate studies (Stewart et al., 2011; Glur et al., 2013; Engeland et al., 2020, Wilhelm et al. 2022) from the Alps or Norway suggest that past flood-rich periods coincided with cool periods. Conversely, climate projections suggest that with global warming, flood occurrence will increase globally and an increase in flood risk is "very likely" in countries representing 70% of the world population (Alfieri et al., 2017; IPCC, 2021). This is because of an increase in heavy precipitation due to increased atmospheric moisture, though changes are region-specific and depend, among other things, on atmospheric circulation changes (IPCC, 2021). Our paper addresses effects of atmospheric circulation changes and of climate warming on European floods on a multidecadal scale, following the work of Blöschl et al. (2020). We applya dynamical perspective to a long period (200 years) that covers both types of flood periods (cold and flood rich, warm and flood rich).

In this paper we specifically explore to what extent atmospheric processes can explain multidecadal variability in flood intensity. We also investigate how the atmospheric contribution can be further partitioned into contributions from circulation changes and moisture changes. To achieve this, we analyze long annual peak streamflow series, daily weather data, reanalyses, and reconstructions.

**2. Data and Methods**

*2.1. Annual peak streamflow series and daily precipitation series*

We use annual maximum streamflow from the Global Runoff Data Center (GRDC) from all series in the region 42-60° N, 2° W to 18° E that are at least 110 years long (in 1904/1905 a network was installed in Switzerland, hence coverage increases; one obviously inhomogeneous series from Sweden was excluded). Note that daily data are not available from this source, hence our focus on annual maximum streamflow. This set was supplemented with two long daily streamflow series from the Rhône (Lang et al., 2016) and Rhine (Wetter et al., 2011), resulting in a set of 45 series (Table S1). For comparison, all series were scaled with their long-term average. The fourteen longest series are shown in Fig. 1a for illustration. For all further analyses, we normalized the series by fitting a Gamma distribution (Botter et al., 2013) and transforming to the quantiles of a standard normal distribution (we also analysed the raw data, which gave similar results). Since in later steps, series will be aggregated, this transformation ensures that combined series have more similar properties. Both the scaling and the transformation to a normal distribution were performed based on a common reference period comprising all data after 1900. We term these series "flood intensity", noting that not each annual value would be called a "flood". For the two daily series, we also analysed the flood frequency (exceedance of the 98[th] percentile, declustered by combining events up to 3 days apart, see Sect. 2.4). A comparison for 30-yr moving averages is shown in Fig. S2. Note that palaeoclimate studies are often based on events with a longer return period (e.g., 10 years or longer; Wilhelm et al., 2021).

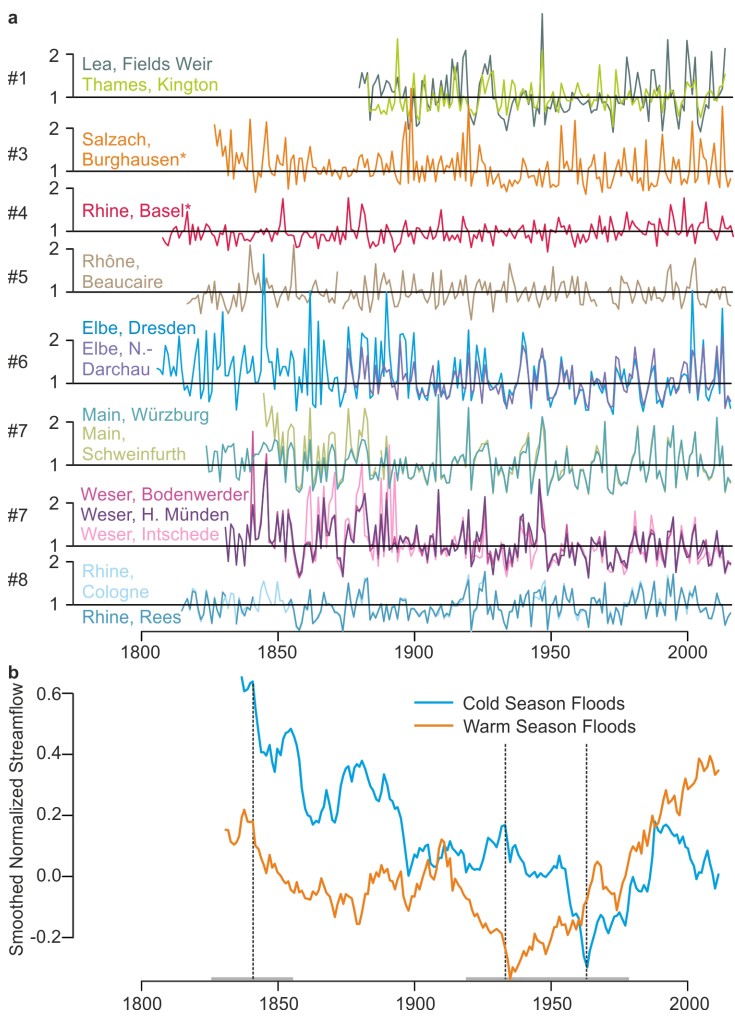

**Figure 1. a** Scaled series of annual peak streamflow for the 14 longest series in Central Europe (Table S1, numbers on the left refer to the regions defined in Sect. 2.2). Stars denote streamflow series with predominantly summer floods. **b** Normalized series of annual peak streamflow averaged (50% of rivers must have data) for rivers with predominantly cold-season floods (blue) and warm-season floods (orange), smoothed with a 30-yr moving average (min. 20 available years)). Dashed lines with grey bars show the 30-yr intervals chosen for analysis.

To each of the streamflow series a daily precipitation record from a neighbouring station was assigned. For this, we searched GHCN daily (Vose et al., 1992), ECAD (Klein Tank et al., 2002) as well as series from MeteoSwiss, and selected series that are as long as possible and, if possible, from a location upstream of the streamflow series (Table S1). Note that in some regions long precipitation records are sparse, and in some cases the same precipitation record was used for more than one streamflow record. Furthermore, it should be noted that these series have not been homogenized and their long-term stability is questionable. Only in one case (Hohenpeissenberg), we accounted for an obvious inhomogeneity by excluding data prior to 1879. From the precipitation series we calculated Rx5day and Rx20day, *i.e.,* the annual maxima of precipitation sum over periods of 5 and 20 days, respectively. The latter is used to characterize the seasonality of hydrological preconditions (e.g., soil saturation) in a catchment, as further discussed in the next section. The former is used as a diagnostic of flood-propelling events. Previous work (Froidevaux et al. 2015, Brönnimann et al. 2019) has shown that flood events are mostly affected by precipitation on 3-4 days prior to the event. Although catchment size varies in our studies, Rx5day is expected to characterize heavy rainfall characteristics over a large range of catchments.

*2.2. Regionalisation*

In a next step, the streamflow series were grouped into regions with hydro-meteorological characteristics as similar as possible using Ward clustering (Ward.D2 in R). We considered the seasonalities of annual maximum streamflow, Rx5day, and Rx20day (i.e., the probability of annual maximum of precipitation over a 5-day window or peak stream flow to fall into a specific month, Fig. S1), the coordinates of the river gauge as well as the coordinates of the precipitation station. The series were standardized and scaled such that streamflow, precipitation, river coordinates, and precipitation coordinates each contributed the same variance. A separation into nine clusters resulted in mostly regionally coherent, non-overlapping clusters. One cluster comprised series from two different catchments (Elbe, Danube) and was correspondingly split and merged with the existing Danube cluster and with an Elbe sub-cluster. Additionally, one river (Ilz) was moved from the Danube cluster (although the Ilz is a tributary of the Danube) to the central Germany cluster as the flood seasonality is clearly distinct from that of the Danube (Fig. S3).

Within the Alpine clusters (Rhône, Alpine Rhine, Danube), individual peak streamflow series show strikingly different trends (Fig. 2). Apart from the fact that the flood season changes from summer (in the Alps) to winter (in the lowland) in all three rivers, which is partly reflected in the clustering as the

change occurs relatively far away from the Alps, also long-term trends radically change from the Alps
to the Alpine foreland. The highest catchments (mean elevation >2000 m asl) in all three regions
(Rhone, Porte-Du-Scex; Rhine Domatems; Inn Martinsbruck) show a strong decrease since the early
20[th] century, whereas the long-term evolution further downstream is flat (Rhône, Chancy) or
increasing (Rhine, Basel; Danube, Achleiten, Fig. 2). A possible explanation relates to the role of
snow processes on high-altitude catchments. Trends could then be due to a superposition of the
seasons of snow melt and heavy precipitation in the early 20[th] century, whereas the two seasons are
more separated today (FOEN, 2021). Other explanations include the role of power plants or other
hydraulic constructions on the flood regime (which is studied for the case of Porte-du-Scex, see
Hingray et al. 2010). In any case, since the focus of this study is on atmospheric processes, these rivers
might confuse our results and hence we removed five series from the three clusters (Inn at
Martinsbruck, Rhône at Porte-Du-Scex, and Rhine at Domatems, Neuhausen, and Rekingen). A one-
series cluster in Sweden (Glomma) also is clearly affected by snow melt and rain-on-snow events (Bøe
et al., 2006). The series are shown in Fig. S4, but not further studied in relation to atmospheric
processes. Our final selection, shown in Fig. 3, comprises a set of 39 streamflow series, aggregated
into eight clusters with areas of ca. 50,000-100,000 km$^2$. The clusters are spatially coherent, internally
consistent with respect to seasonality and heavy precipitation regime, and internally homogeneous
with respect to time evolution (exceptions are Southern England, where the only two long series
disagreed, and the Danube, where time evolution is less homogeneous). The clusters represent
Southern England, Southern Norway, the Rhône, the Alpine Rhine, the Lower Rhine, Central
Germany, the Elbe, and the Danube.
Seasonality is an important factor to consider as it is characteristic for a given region. Furthermore, the
relevance of atmospheric process changes in the course of the year. Winter events tend to be related to
different circulation patterns (e.g., zonal flow) than summer events (Stucki et al. 2021). Moreover, the
role of convection is stronger in summer. In the following we will therefore perform all analyses for
annual data as well as for annual series restricted to flood seasons, defined as May ro October (for
clusters Upper Rhine and Danube) and November to April (all other clusters). This partitioning
captures the seasonal flood characteristics as well as the seasonal differences in atmospheric processes
and it still ensures an adequate sample size.

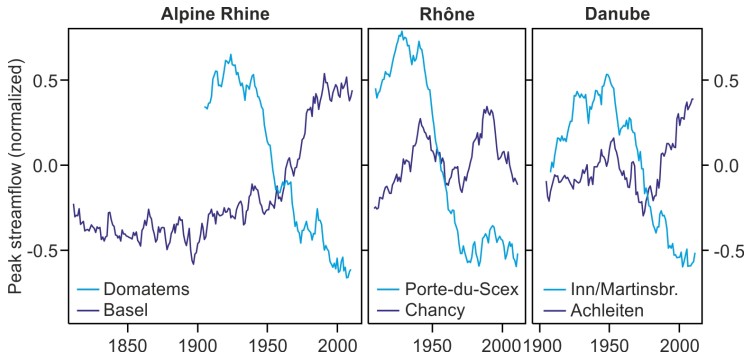

**Fig. 2.** Normalized smoothed streamflow series for the three Alpine regions. In each region an upstream catchment (mean altitude >2000 m asl, light blue) and streamflow series downstream from the same river system (dark blue) is shown. All series are smoothed with a 30-yr moving average.

*2.3. Atmospheric and climate data*

The focus of the paper is on the atmospheric contribution to flood intensity. However, studying atmospheric circulation 200 years back in time with a focus on extreme weather events is challenging. To compensate for potential deficiencies of long-term data sets and to obtain more robust results, we use multiple atmospheric data sets that are partly independent and are based on different methods.

The dynamical reanalysis 20CRv3 (Slivinski et al., 2019) provides 3-hourly, 3-dimensional, global atmospheric data back to 1806. 20CRv3 assimilates only surface pressure observations into an atmospheric model with prescribed sea-surface temperatures, sea-ice concentration, and radiative forcings. It consists of 80 equally likely members. All analyses shown here were performed for each member to obtain a physically plausible range of realisations. We extracted one grid point per region (crosses in Fig. 4; selected from the 1x1° grid such as to best represent atmospheric processes relevant for the region; note that we preferred point data, as the Rx5day data also are point data). The reanalysis allows calculating specific diagnostics, such as moisture flux convergence, at a relatively high resolution. However, the quality of 20CRv3 varies in time and space, particularly during the 19[th] century. The data prior to 1836 are less well evaluated and have a larger uncertainty (Slivinski et al., 2021). We always show the ensemble mean and ±1 ensemble standard deviations.

The second data set consists of daily weather types. Floods occur during specific weather patterns with similar hydro-meteorological characteristics (Stucki et al., 2012) and thus weather type classifications can be useful to study atmospheric contributions to floods. We use the Swiss CAP7 weather types back to 1763 (Cluster Analysis of Principal Components, Schwander et al., 2017) which is based on daily meteorological data from Europe, some of which overlap with 20CRv3.

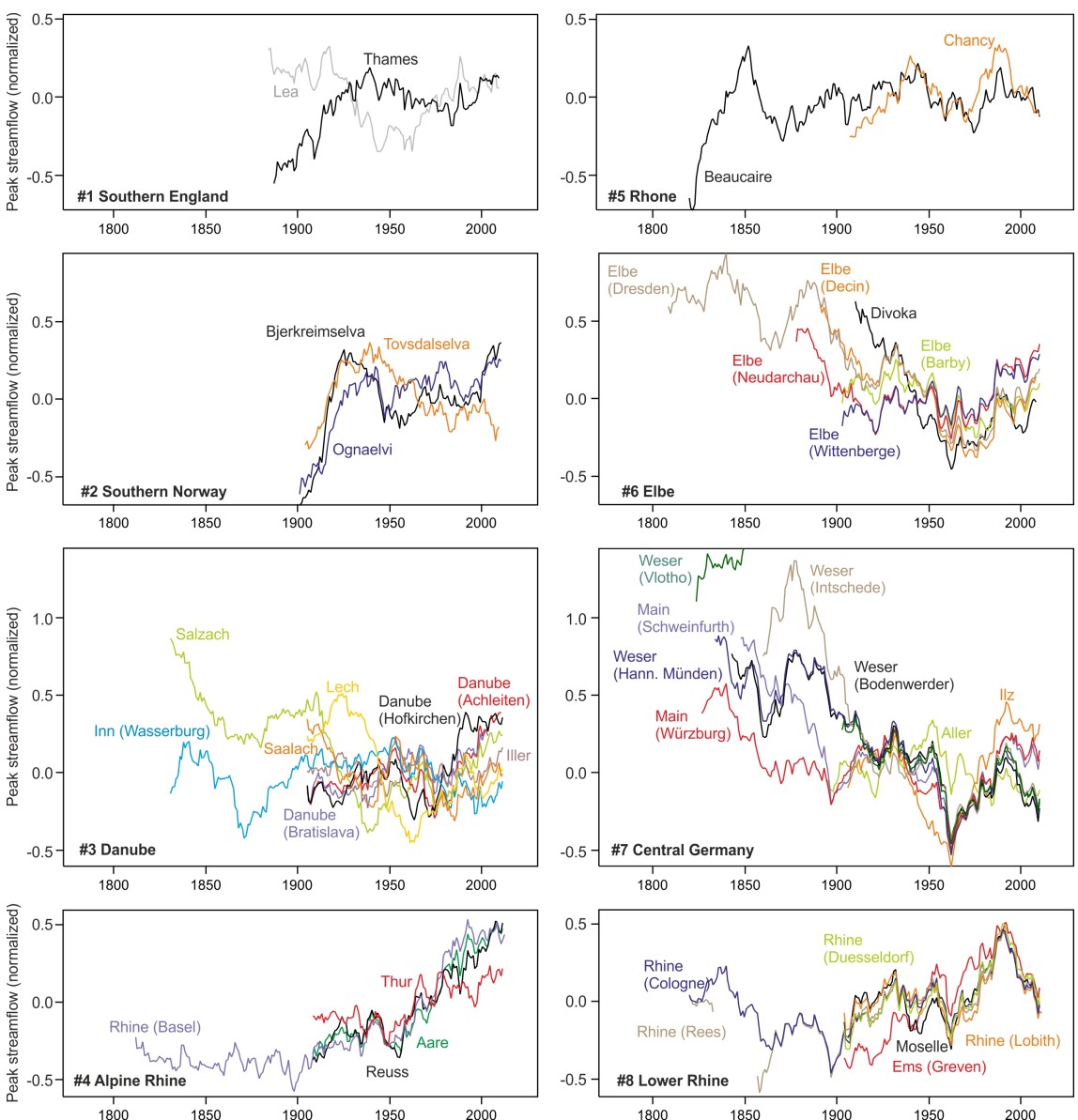

**Fig. 3.** Normalized smoothed streamflow series for all series in all eight clusters. All series are smoothed with a 30-yr moving average.

The third data set is the updated global atmospheric paleo-reanalysis EKF400v2 covering the last 400 years (Franke et al., 2020; Valler et al., 2021). EKF400v2 provides monthly global 3-dimensional reconstructions from an offline assimilation. While there is a small overlap in input data with 20CRv3 (some of the pressure series), EKF400v2 mainly assimilates other data (temperature, precipitation, documentary data, tree-rings). However, unlike for the other two data sets, EKF400v2 is not available at daily resolution. We use the monthly values to analyse seasonal precipitation and 500 hPa geopotential height (GPH).

For comparison with climate model data, we analyse monthly precipitation also directly in station data (Peterson and Vose, 1997; Alexander and Jones, 2001; Murphy et al., 2018) and in the observation-based gridded product HISTALP (Efthymiadis et al., 2006), which also includes temperature (note that these data were assimilated into EKF400v2).

194

*2.4. Flood probability index*

Based on the weather types, we define a Flood Probability Index (FPI see below), which characterizes a season or year based on sequences of weather types. To calibrate the index we need daily streamflow series, which are available only for the Rhine (Basel) and Rhône (Beaucaire). We calculate it separately for the warm season (May to October, for Basel) and cold season (November to April, Beaucaire) in order to analyse the seasonally-varying relation of weather types with temperature anomalies. The calculation of the FPI is based on Quinn and Wilby (2013) and is performed exactly as in Brönnimann et al (2019). We first determined the 98[th] percentile of daily streamflow within the respective seasonal window and marked all days above this percentile as extreme events. Events separated by 3 or fewer days were combined to ensure independence, and from each sequence of marked days only the day of the maximum was kept. For each weather type we then calculated the fraction of days coinciding with a flood event relative to all days of that type. Then we assigned this number to each day of that weather type. This was repeated for different lead times up to 5 days such that the weather on preceding days is also considered, and lead times 5 to 0 were weighted 1/16, 1/8, 3/16, 1/4, 1/4, and 1/8. This window length and weighting was taken from a previous study (Brönnimann et al., 2019) and was based on analyses of daily discharge, precipitation, and water flux convergence on the preceding days. This procedure yields an FPI for each day in the past (note that the index was calibrated in the data after 1900, but calculated back to 1763). The 75[th] percentile of this index calculated for each season was then chosen as an indicator of flood probability (for details see Brönnimann et al., 2019).

215

*2.6. Water flux convergence*

Atmospheric circulation was furthermore analysed in terms of advection and convection of moist air. We calculated a simplified measure of moisture flux convergence in which 850 hPa horizontal wind is multiplied with precipitable water, termed water flux convergence in the following. This was calculated for each of the 80 ensemble members of 20CRv3 and each 3-hour interval. In this analysis we use the annual maximum 5-day average, CONV5d (analog to Rx5day; different windows from 3 hours to 10 days gave very similar results). All series were smoothed with a 30-year moving average and finally the members were averaged. CONV5d indicates intense moisture transport and precipitation.

Based on the 3-hourly values feeding in to the maximum 5-day value, we decomposed CONV5d into its contributions as follows (overbar denotes the average over the entire period (1806-2015), primes denote deviations therefrom, $q$ denotes precipitable water, $\vec{v}$ is the wind vector):

$$-\vec{v} \cdot \left( \left( \overline{q} + q' \right) \cdot \left( \overline{\vec{v}} + \vec{v}' \right) \right) =$$

$$-\overline{q} \cdot \left( \frac{\partial \overline{u}}{\partial x} + \frac{\partial \overline{v}}{\partial y} \right) - \overline{u} \cdot \frac{\partial \overline{q}}{\partial x} - \overline{v} \cdot \frac{\partial \overline{q}}{\partial y}$$

$$-\overline{q} \cdot \left( \frac{\partial u'}{\partial x} + \frac{\partial v'}{\partial y} \right) - u' \cdot \frac{\partial \overline{q}}{\partial x} - v' \cdot \frac{\partial \overline{q}}{\partial y}$$

$$-q' \cdot \left( \frac{\partial \overline{u}}{\partial x} + \frac{\partial \overline{v}}{\partial y} \right) - \overline{u} \cdot \frac{\partial q'}{\partial x} - \overline{v} \cdot \frac{\partial q'}{\partial y}$$

$$-q' \cdot \left( \frac{\partial u'}{\partial x} + \frac{\partial v'}{\partial y} \right) - u' \cdot \frac{\partial q'}{\partial x} - v' \cdot \frac{\partial q'}{\partial y}$$


This decomposition results in four groups of three terms. The first three terms on the right hand side
(second line) indicate the contribution by the mean flow, the next three terms (third line) the
contribution by changes in circulation (while keeping moisture constant), the next three terms measure
the contribution by changes in precipitable water (while keeping the circulation constant) and the last
three terms describe the interaction of circulation and moisture changes.

*2.7. Model simulations*
To test the effect of sea-surface temperature and external forcing on multidecadal variations of
atmospheric circulation, we used the global atmospheric model ECHAM6 (Giorgetta et al., 2013). It
was run in the standard configuration T63L47 for the years 1851-2015. The spatial resolution
corresponds to ca. 1.9°. In total 31 members were produced using different initial conditions as well as
different sea-surface temperatures (obtained by sampling from the ten members in HadISST2); only
one realization was available for sea ice (Titchner and Rayner, 2014). All other forcings (land surface,
volcanic aerosols, tropospheric aerosols, and greenhouse gas concentrations) followed the
Paleoclimate Modelling Intercomparison Project (PMIP) protocol (Jungclaus et al., 2017). Ensembles
with individual forcings are not available.

**3. Results and Discussion**
*3.1. Annual peak streamflow*
The longest 14 series show that extreme floods occurred in the 19[th] century, particularly in the Elbe,
Weser, and Main catchments, but also Salzach and Rhône show high peaks. Conversely, apart from
floods in 1946 (Weser) and 1947 (Lea, Thames, Main), the period ca. 1940 to 1970 exhibits fewer
spikes. However, the rivers exhibit different streamflow regimes and flood seasonalities (Fig. S2). The
upper (Alpine) catchments of Rhine and Danube exhibit their annual maximum streamflow typically
during the warm season, most other catchments during the cold season. After normalizing, the "cold
season" and "warm season" rivers were therefore averaged separately and the series were smoothed in
Fig. 1b. Likewise, all further analyses were performed for annual series as well as for flood seasons
(i.e., Nov-Apr for "cold season" flood rivers and May-Oct for "warm season" flood rivers). Note that
throughout the paper, a 30-yr moving average was used for visualisation, where at least 20 values must
be available. For averaging regions we require that half of the regions have available data; only when
averaging within regions we did not require a minimum as the chosen clusters were largely
homogeneous such that the drop-out of a series will not have a large effect.
These aggregated curves show additional features. Less pronounced peaks for cold-season flood rivers
are found in the 1870s and the early 20[th] century. Based on peaks on the cold-season series, three 30-yr
periods were selected for further investigation: 1827-1856 (primary maximum), 1949-1978 (primary
minimum), and 1919-1948 (local maximum at a time when warm-season series exhibit low values).
While numerous non-climatic factors (e.g., changes in the stream network and land use) contribute to
long term trends or may induce step changes (e.g., Hingray et al. 2010), multidecadal variability is less
influenced by such changes (note that the Rhine series was corrected for two such changes) and hence
climatic conditions are analysed.
Our findings of increased flood intensities in Central Europe in the 19[th] century and a decrease in the
mid-20[th] century are confirmed by documentary evidence (Naulet et al., 2005; Wetter et al., 2011;
Himmelsbach et al., 2015; Lang et al., 2016). A recent, comprehensive study based on documentary
data and a three-class flood magnitude index (Blöschl et al., 2020) found coherent flood phases in the
mid-19[th] century in Central and Southern Europe, in the early 20[th] century in northwestern Europe, and
in recent decades in Central and Western Europe, although this is not the case for each individual river
(Glaser et al., 2010).
Our aggregation into eight regions retains the main phases of flood intensity but adds spatial
information. This is shown for annual time series (Fig. S4) as well as for flood seasons (Fig. 4). High
peak streamflow occurred in Central Europe in the 19[th] century, in Central and Western Europe in the
early 20[th] century, low peak streamflow in all regions after 1950. Since 1970 peak streamflow has
increased, although not everywhere, and some series (not only those influenced by snow) show a
decline at the beginning of the 21[st] century.
For comparison with Blöschl et al. (2020), we add the interpolated and smoothed series calculated
from their data and code to Fig. S5. Correlations (at 4-yr aggregation, corresponding to the voxel size
in Blöschl et al. (2020)) with peak streamflow (numbers in Fig. 4) are around or below 0.4,
statistically significant (t-test, $p < 0.05$) for the regions Southern Norway, Upper Rhine, Rhone, and
Elbe. Obviously, the comparability of measurement-based versus document-based evidence is limited.
For instance, analysed statistics differ (annual maxima versus indexed extremes), the series measure
different aspects of flood (streamflow versus documented flood intensity) and there is large river-to-
river variability. Yet, the flood-rich decades in the middle and late 19[th] century in Central Europe, in
the early 20[th] century in Northwestern Europe, the Europe-wide flood-poor period after 1950, and the
recent increase in flood intensity are salient features of all analyses. This becomes clear when
aggregating the series spatially into Northwestern Europe (UK and Southern Norway) and Central
Europe (all other regions) and smoothing the Blöschl data for better comparability with the 30-yr
smoothed streamflow (see Fig. S5). Hence, the regional characteristics are consistent with the
documentary evidence on a climatological scale, and the fact that corresponding periods of more and
less frequent floods are found with both methods opens the door for the following analyses.
In the following, we show results only for the seasonal series (results for the annual series are similar).
Note that flood seasons capture ca. 80% of peak streamflow events, and flood intensities are ca. 8%
higher than on out-of-season floods.

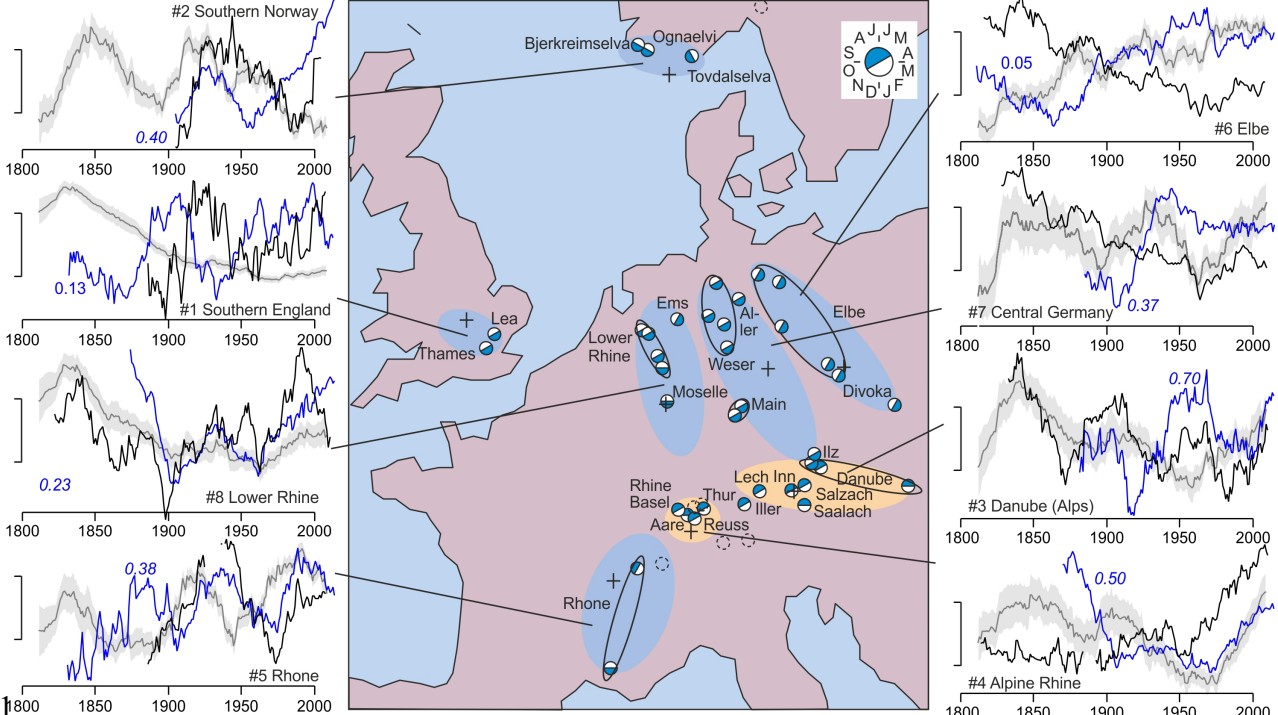

**Figure 4.** Regionally averaged (coloured ellipses; black ellipses indicate same river) series of normalized peak
streamflow (black), Rx5day (blue, the number indicates its correlations with peak streamflow at 4-yr aggregation,
italics indicates p<0.05) and CONV5d during the flood season from 20CRv3 at locations of crosses (grey, shading
indicates the ensemble standard deviation), standardized and subsequently smoothed with a 30-yr moving
average (scale bars range from -0.5 to +0.5). Regions are colour-coded according to the predominance of cold
(blue; Nov-Apr) or warm season floods (orange; May-Oct). The blue part of the white-blue circle for each river
indicates the 6-month period with highest flood frequency). Dashed circles: Streamflow series excluded because
of likely influence of snow melt, or hydropower dams or other hydraulic constructions on trends.
*3.2. Atmospheric influences and the role of circulation and water vapour changes*
First, we analysed the relation between flood intensity and precipitation. In most regions, flood
intensities are statistically related to Rx5day. Correlations (Figa. 4 and S5, calculated from annual
data) vary greatly (between 0.05 and 0.7), but are significant (t-test, p<0.05) for six regions. Note that
a high correlation is not necessarily expected on a year-to-year scale as Rx5day events often do not
occur together with annual peak streamflow. In winter flood regions, for instance, Rx5day occurs
predominantly in summer, whereas peak streamflow occurs predominantly in winter, hence a winter
series is correlated with a summer series. Nevertheless, years with high peak stream flow coincide
with years with high maximum 5-day precipitation, although the association is not very strong and one
needs to keep in mind that flood intensity is not purely atmospherically driven. Note also that neither
peak stream flow (except for Rhine, Basel) nor Rx5day are based on homogenised data series.
Next, we analysed atmospheric influences on the multidecadal variability of peak stream flow using
the diagnostics defined in Sect. 2. The CONV5d series (grey lines and shading in Fig. 4; for
visualization they were standardized prior to filtering) exhibit multidecadal variations with maximum
convergence in the 19th and early 20th century and minimum convergence around 1950, although the
pattern differs from region to region. They are in general agreement with the maximum streamflow
curves for several regions (e.g., Rhône, Lower Rhine, Central Germany, Danube), while in other
regions the agreement is worse. Similarly as for Rx5day, CONV5d is less reliable in the early years,
prior to ca. 1836. The steep increase in these years therefore cannot be assessed.

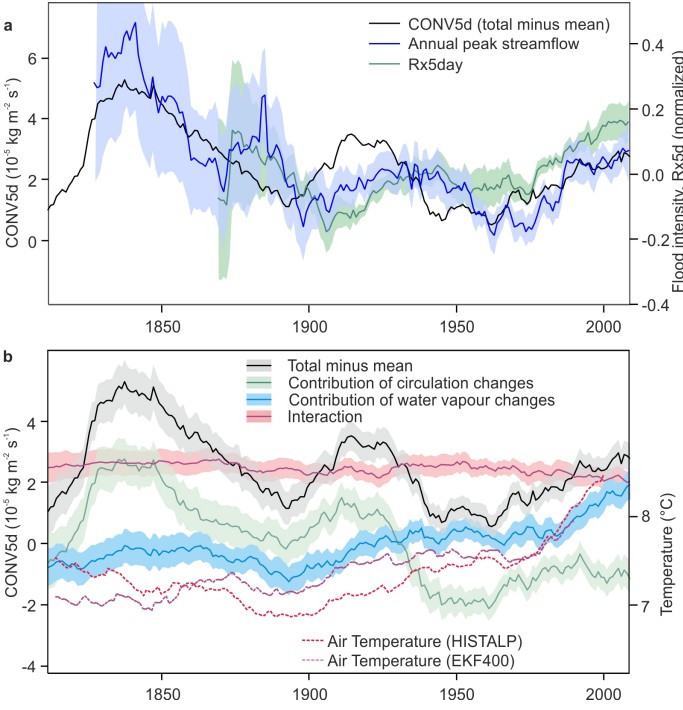


**Figure 5. a** Average of regional averages of annual maxima of peak streamflow, Rx5day, and CONV5d. Shading
indicates ±1 standard error. **b** Contributions to CONV5d from circulation changes, water vapour changes, and
their interaction. Shading indicates ±1 standard deviation of the ensemble. Dashed curves show annual mean
temperature from HISTALP and EKF400. All curves are smoothed with a 30-yr moving average.
While all individual indicators (flood intensity, Rx5day, CONV5d) have uncertainties that are
particularly large in the early decades, there are also clear similarities. A further aggregation reveals
the common low-frequency variability even more distinctly. When averaging all three indicators
across all eight regions (Fig. 5), we find a close similarity after around 1870. All series show the

recent increase, the minimum in the 1960s, a peak around the 1930s, and a minimum around 1900, as already noted in Fig. 1. Flood intensity and CONV5d also show a peak in the 1840s, which is however not seen in the (sparse) Rx5day data. The association between the three series is further supported by cross-wavelet analyses (Fig. S6), which shows significant relations at time scales longer than ca. 30 years.

Thus, despite the uncertainties, we can use these indicators to trace the atmospheric impacts on the multidecadal variability in flood intensity. The atmospheric processes, in turn, can be partitioned into contributing processes as described in Sect. 2. Figure 5b shows the contributions from circulation changes, from water vapour changes, and from their interaction. The interaction term is negative with only small changes over time. The contribution from circulation changes (green line) dominates and shows all main features found in CONV5d. However, the long term trend differs. This is due to changes in water vapour (blue line). The contribution of water vapour changes shows a two-step increase after 1900.

An analysis of linear trends in the unsmoothed series since 1963, the minimum in flood intensity, reveals an increase in CONV5d ($4.04 \times 10^{-7}$ kg m$^{-2}$ s$^{-1}$ yr$^{-1}$, which is not statistically significant), no trend in the contribution of atmospheric circulation changes, but a highly significant increase in the contribution of water vapour changes ($6.13 \times 10^{-7}$ kg m$^{-2}$ s$^{-1}$ yr$^{-1}$). The contribution of water vapour changes depends on temperature through the Clausius-Clapeyron relation. To illustrate this relation, annual mean temperature in HISTALP (Efthymiadis et al 2006), the longest gridded observational data set, and in EKF400v2 for the same regions are plotted such that 1 °C corresponds to 0.46 $10^{-5}$ kg m$^{-2}$ s$^{-1}$. This is equivalent to a 6.5% change in CONV5d, the number expected following the Clausius-Clapeyron relation if annual maxima would follow the annual average trend (saturation can be assumed for annual maximum moisture convergence). After around 1900, the general pattern and amplitude of the contribution of water vapour changes is consistent with an increased intensity of heavy precipitation in a warming atmosphere, although the amplitude of the CONV5d increase is somewhat smaller than that of the scaled temperature increase.

In fact, this might help to explain the varying relation between temperature and floods over time: Palaeoclimate studies (Stewart et al 2011, Glur et al 2013, Wilhelm et al. 2021), particularly from the northern Alps, suggest that past flood-rich periods coincided with cool periods, while climate projections suggest that with global warming, flood occurrence may increase in certain regions. Although palaeoclimate studies often are based on small catchments, target a longer return period and a low-frequency variability scale that is longer than decades as in this study, it is nevertheless interesting to analyse the relation between temperature and floods on a multidecadal scale.

To analyse the role of circulation for temperature, we used the FPI index for the Rhône and Rhine, which was calculated specifically for the corresponding flood seasons (Nov-Apr for the Rhône, May-Oct for the Rhine). This index measures the frequency of flood-prone weather types, to which cyclonic

weather types contribute very strongly. As a consistency test, the smoothed curves (Fig. 6a) show high values in the 19th and early 20th century and a decrease after ca. 1950; further analyses of the FPI index for Basel are shown in Brönnimann et al. (2019). For the following analysis we used the unsmoothed, but detrended FPI indices, onto which we regressed the detrended temperature fields of the corresponding seasons (Fig. 6b). For the Rhine, which is mostly affected by summer floods, flood prone seasons are typically cold. Conversely, for the Rhône, with typically winter floods, flood-prone seasons are warmer than average in the lowland, but colder than average at higher altitudes. Both are consistent with a predominance of cyclonic weather types over Switzerland: They bring colder than average weather in summer, but warmer than average in winter except at high altitudes, which normally, but not during cyclonic weather types, are above an inversion.

This means that from the contribution of circulation alone, flood-rich periods in summer-flood regions and generally in the Alps are expected to be cool. This is not the case after 1980, when the partitioning (Fig. 5b) shows a growing contribution of water vapour increase whereas the contribution of circulation changes is constant (and the FPI is low, Fig. 6a). Warming phases (also in the past) rather directly lead to an increase in CONV5d, but warming may be driven by atmospheric circulation changes that decrease CONV5d, or it may be driven by other forcings in which case atmospheric circulation does not counteract the increase in CONV5d.

*3.4. Regional differences in circulation effects*

Circulation changes had regionally different imprints in different times. Recall that 1827-1856 was flood-rich in Central Europe (year-round), 1919-1948 was flood-rich in northern and western Europe (cold season), 1949-1978 was flood-poor across Europe (year-round, Fig. 1). The contribution of circulation changes to CONV5d (shown in Fig. 7 for each region) is consistent with this result. Some regions show an almost opposite behaviour to each other. For instance, in the mid 19th century, circulation changes contributed to high CONV5d in Southern Norway but to relatively low values in the Rhône catchment, whereas the opposite was the case in the second half of the 20th century (Fig. 7). While the contribution of circulation differs from region to region, the contribution from water vapour changes is more uniform and shows an increase in all regions.

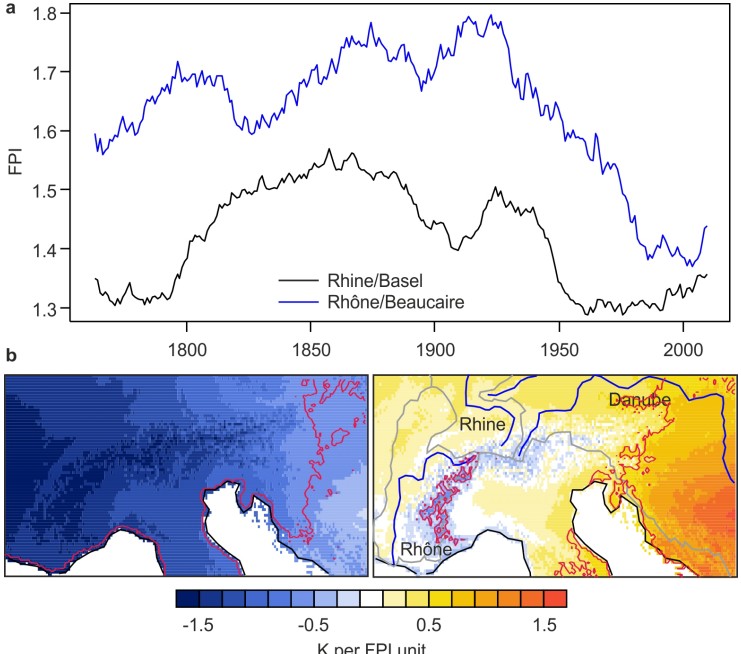

**Figure 6. a.** FPI index for the Rhine in Basel (May-Oct) and the Rhône in Beaucaire (Nov-Apr), smoothed with a 30-yr moving average. **b.** Regression map of detrended seasonal (May-Oct and Feb-Apr, respectively) mean temperature in HISTALP onto the corresponding (detrended) FPI indices. Red lines indicate significant (p<0.05) coefficients.

To test whether these spatial differences due to atmospheric circulation are reflected in the seasonal mean large-scale flow, we analysed (Fig. 8) 30-yr averages of seasonal mean anomalies in precipitation and 500 hPa GPH in EKF400v2 and observations (Peterson and Vose, 1997; Alexander et al., 2001; Murphy et al., 2018). In terms of seasonal mean precipitation, the cold seasons 1827-1856 and 1949-1978 show a rather mixed signal. Although not inconsistent with the observed multidecadal flood intensity, one would probably not address these periods as flood-rich and flood-poor, respectively, based only on seasonal mean precipitation (note that Blöschl et al. (2020) define a flood period in 1840-1872; corresponding plots exhibit similar patterns as for 1827-1856; Fig. S7).

The period 1827-1856 (cold season) shows a pressure pattern that is similar to a negative mode of the North Atlantic Oscillation or East Atlantic Pattern, but with the positive pressure anomaly displaced southeast of Iceland. Seasonal mean precipitation (both in EKF400v2 and station data) shows a mixed signal; with slight increases in the Rhône catchment, Central Europe, and Southern Norway, but drying over England. The warm season show negative anomalies of 500 hPa GPH over the entire continent, accompanied by increased rainfall, which is consistent with frequent flood-prone weather.

The 1919-1948 cold season average shows negative 500 hPa GPH anomalies over the Atlantic and increased precipitation over Western Europe, which agrees with the increased flood intensity in this region. The clearest signal is found for the flood-poor period 1949-1978 in the warm season. The analysis show pronounced drying and positive anomalies of 500 hPa GPH. The start of this period, which coincided with massive droughts (*e.g.,* Brazdil et al., 2016) was accompanied by a poleward shifted subtropical jet (Brönnimann et al., 2015).

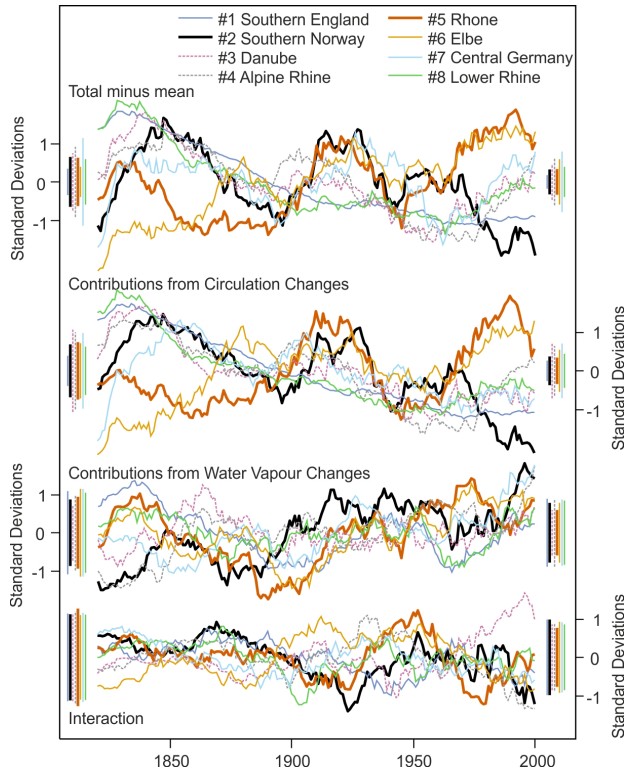


**Figure 7.** CONV5d (total minus mean) and contributions to it from circulation changes, water vapour changes,
and their interaction for each of the eight regions (ensemble mean). All series were standardized and smoothed
with a 30-yr moving average. Coloured bars indicate ±1 one ensemble standard deviation at the beginning and
end of the period (the change inbetween is close to linear).
We further addressed the underlying causes of multidecadal anomalies by analysing, in the same way
as EKF400v2, an ensemble of 31 simulations with the ECHAM6 atmospheric model starting in 1851
(the 1827-1856 period cannot be analysed). The precipitation anomalies and the broad features of
GPH anomalies found in EKF400v2 are rather well reproduced for the 1919-1948 and 1949-1978
periods, both cold and warm seasons (for 1840-1872 see Fig. S5). For instance, for the cold season, the
negative GPH anomalies over the North Atlantic in 1918-1948 and the zonal pattern of low GPH over
the eastern North Atlantic and high GPH over Russia in 1949-1978 agree well. The wet conditions in
western Europe in 1919-1948 in winter and the dry conditions in 1949-1978 in summer are highly
significant in the atmospheric model simulations. The latter is arguably the most significant feature in
the model analysis. Although this analysis concerns only changes in the seasonal means, not in
extremes, it shows that atmospheric model simulations forced with, among other factors, sea-surface
temperatures are able to reproduce some characteristic features of atmospheric circulation changes.
However, the seasonal mean circulation and precipitation describes the flood conditions only to a
limited extent (see Zanchettin et al., 2019, for the role of Atlantic sea-surface temperature variability
for floods). Note, also, that also EKF400v2, despite the large number of observations assimilated, is
dependent on sea-surface temperature input to the underlying model. Overall, the model simulations
suggest that part of the multidecadal variability can be reproduced from model boundary conditions
(note that in addition to sea-surface temperature, they also encompass external forcings).

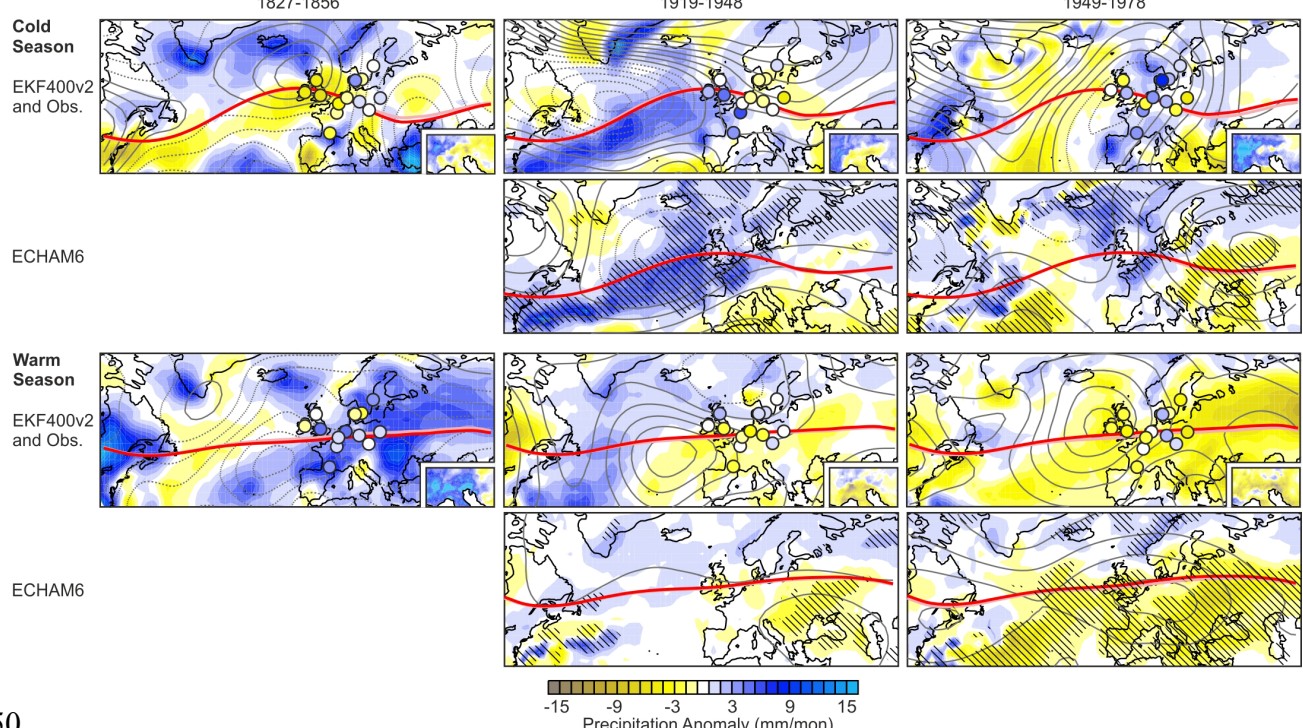

**Figure 8.** Simulated atmospheric circulation and precipitation. Anomalies (with respect to 1851-1950) of precipitation (colours) and 500 hPa GPH (contour distance 2 gpm centered around zero, dashed contours indicate negative numbers) in the 30-yr periods 1827-1856, 1919-1948, and 1949-1978 in the EKF400v2 reconstruction (ensemble mean), observations (insets: HISTALP; circles: GHCN), and ECHAM6 simulations (hatching denotes 95% significance of precipitation anomalies, calculated from the 30-year averages of the 31 members using a one-sample t-test). Thick red lines show the GPH contour 5450 gpm (cold season) or 5650 gpm (warm season; light pink: same for 1851-1950).

## 5. Conclusions

Long time series of annual peak streamflow in Western and Central Europe exhibit substantial multidecadal variability, consistent with previous work by other authors. Flood-rich phases occurred in the 19[th] century in several regions, in the early 20[th] century in western and northern Europe, and since the 1980s, while a flood-poor period occurred after the second world war. The flood variability is in line with observed changes in Rx5day (except in the mid-19[th] century, which however could be due to a lower data quality).

Annual peak atmospheric water flux convergence in a reanalysis also shows the same pattern of multidecadal variability as flood intensity and Rx5day, and this is further supported by an indicator based on weather types. Although the uncertainties in each data set are large, results are robust and show the same main phases of low-frequency variability. The reanalysis data allow a more physical interpretation. Partitioning the atmospheric water flux convergence into contributions from circulation and water vapour changes, we find that peak streamflow of European rivers from around 1820 to 1980 was largely forced by atmospheric circulation changes. In contrast, the recent increase in moisture flux

convergence was to a larger part driven by increasing atmospheric moisture due to climate change.
This might contribute to explaining why in the past, flood-rich periods coincided with cold periods
(particularly in summer-flood regions such as the northern Alps, to which many proxy studies refer)
while more floods may be possible in Europe in a future, warming climate. Note, however, that
paleoclimatic studies often address longer time scales, smaller catchments, and longer return periods
than are used in this study.
Changes in seasonal mean atmospheric circulation partly mirror the changes in flood intensity
changes. Important features of these changes are reproduced in atmospheric model simulations,
indicating that oceanic forcing might play a role. This is specifically the case for the dry and flood-
poor summers 1949-1978.
The thermodynamic effect is likely to increase further. The floodings in Central and Western Europe
the summer of 2021 fit into the picture of a stronger thermodynamic contribution. However, flood
projections in Europe under different emission scenarios remain unclear (Kundzewicz et al., 2017), as
several sources of uncertainties have to be considered (climate models, downscaling, hydrological
models) and projections for flood intensity (e.g. Roudier et al., 2016), frequency (e.g. Giuntoli et al.,
2015) or both (e.g. Alfieri et al., 2015) in European rivers vary.

*Acknowledgements:* This work was supported by Swiss National Science Foundation project WeaR (188701),
and by the European Commission (ERC Grant PALAEO-RA, 787574). Simulations were performed at the Swiss
National Supercomputing Centre CSCS. Support for the Twentieth Century Reanalysis Project version 3 dataset
is provided by the U.S. Department of Energy, Office of Science Biological and Environmental Research (BER),
by the National Oceanic and Atmospheric Administration Climate Program Office, and by the NOAA Physical
Sciences Laboratory. We acknowledge the data providers in the ECA&D project.

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

**Data availability**
The GRDC data can be downloaded here: https://www.bafg.de/GRDC/EN/Home/homepage_node.html
Flood series on the Rhône river at Beaucaire (1816-2016) is available from: https://www.plan-
Rhône.fr/publications-131/actualisation-de-lhydrologie-des-crues-du-Rhône-
1865.html?cHash=5628938abe287dc9ca390dad7373ae0e
EKF400v2.0 is available from: https://doi.org/10.26050/WDCC/EKF400_v2.0, 2020
20CRv3 is available here: https://portalnersc.gov/project/20C_Reanalysis/
HISTALP is available here: http://www.zamg.ac.at/histalp/datasets.php
The CAP7 weather types are available from https://cp.copernicus.org/articles/15/1395/2019/, the Lamb weather
types are available from https://doi.pangaea.de/10.1594/PANGAEA.896307
**Code availability**
The code for the processing of the streamflow data as well as for generating the FPI is attached as supplementary
file together with all input data.
**Author contributions**
SB designed the studies and did most of the analyses and writing. PS processed reanalysis data, JF, VV, and YB
provided the EKF400v2 data and helped in the analysis, RH performed the climate model simulations, LCS,
GPC and PDS provided the 20CRv3 reanalysis data and interpretation, ML provided the Rhône data and BS
assisted in the hydrological analyses. ML and BS assisted in the hydrological interpretations. All authors actively
discussed the results and all authors contributed to writing.