# Peer review of "Influence of Warming and Atmospheric Circulation Changes on"

_Climate of the Past, 2021_

## Author Response (AR1)

**Reply to reviewer 1**

This study uses streamflow, daily weather data, reanalyses, and reconstructions to explore variations in European flood frequency during the past 2 centuries and reconcile indications of European flood frequency to be higher under warm as well as cold background climate conditions. The study illustrates how the relative role of atmospheric circulation and moisture content for moisture flux convergence, hence precipitation, changed historically. Based on the moisture content contribution becoming predominant in recent, warmer times, the manuscript discusses implications of such findings for projected floods.

I found the study overall well-conceived and the manuscript well organized and well written. I especially appreciated the efforts to combine different sources of information including observations, model output and reconstructions. I found the analysis overall sound and the conclusions well supported by the results. However, I have a few comments on the study that I ask the Authors to consider in a revised version of the manuscript.

On a general note, I would appreciate a stronger focus on the statistical analyses supporting the existence of linkages between the considered processes, for instance in terms of significance of co-variability between time series. I provide a few specific comments below to illustrate the occasions when I felt the interpretation of results requires further support. Similarly, the comparison between information from different sources appears occasionally to be only qualitative. This left me wondering about the purpose of some of the comparative analyses provided in the manuscript: central or just ancillary to show uncertainty? I think the manuscript would benefit from a bit more guidance by the authors about the purpose (and expected outcome) of some of the analyses. Again, I highlight the few occasions when this occurred in the specific comments below.

Concerning the adopted methodology, the only main question I have regards the normalization: If I understand the method correctly, normalization is over the whole length of a time series. The relative representation of trends is affected by the fact that series of different length are compared (those including the trend period and those extending further back, during period of little or no trend). If this is true, maybe a word of caution on this approach when comparing trends in figures 2 and 3 or when different discharge time series are averaged out in Figure 1b can be appropriate. An alternative approach could be to normalize over periods as similar as possible across the different series (for instance using 1900-2000). A few more specific methodological questions follows in the specific comments.

Thanks for this comment. Indeed, the normalization was done over the maximum possible length. In the revised manuscript we will use a normalisation on all data after 1900. Results are very similar

Specific comments

Section 2.1: The authors use annual maximum streamflow as a reference for their analysis of floods. What if more floods occur within one year? In my understanding this possibility is not accounted for in the analysis, but might be relevant for the overall assessment of flooding statistics. In section 2.4, it comes clear that daily streamflow series are available for only two stations, so I guess this aspect is difficult be assessed. Nonetheless I feel some discussion in section 2.1 would be worth it.

The reviewer is correct that we have only two series with daily data, which allow addressing that question. We now discuss the question of flood definition more prominently in Sect. 2.1 and added, for these two series, a plot of flood frequency (number of events exceeding the

98$^{th}$ percentile, declustered) to the Supplement (Fig. S2). The general results are similar, a sentence is added:

*"For the two daily series, we also analysed the flood frequency (exceedance of the 98$^{th}$ percentile, declustered by combining events up to 3 days apart, see Sect. 2.4). A comparison for 30-yr moving averages is shown in Fig. S2."*

Line 89-90: "From the precipitation series we calculated Rx5d and Rx20d, *i.e.,* the annual maxima of precipitation sum over periods of 5 and 20 days, respectively." Is the temporal connection with the flood event checked? As far as I see it, especially for "flood intensities" that are about average this may not be reflective of a true connection between precipitation and discharge.

In our paper, Rx20day is used to characterise catchments hydrologically mainly with respect to seasonality. For typical river floods, shorter periods are relevant. We have checked this for one station (Basel) in a previous publication (Brönnimann et al., 2019) and found that the 2-3 days prior to the event are the most relevant (5 days prior to the event precipitation is already above the 75$^{th}$ percentile, but this is not extreme). There is a more systematic study on this by Froidevaux et al. (2015) concluding "that the consideration of a 3–4 days precipitation period should be sufficient to represent (understand, reconstruct, model, project) Swiss Alpine floods." Note that the size of catchments varies largely in our study; some are larger than those studied in Froidevaux et al., some are of similar size. Eventually we aggregate series regionally (both precipitation index and streamflow). Hence, Rx5day should be a good choice.

We discuss in the revised manuscript:

*"From the precipitation series we calculated Rx5day and Rx20day, i.e., the annual maxima of precipitation sum over periods of 5 and 20 days, respectively. The latter is used to characterize the seasonality of hydrological preconditions (e.g., soil saturation) in a catchment, as further discussed in the next section. The former is used as a diagnostic of flood-propelling events. Previous work (Froidevaux et al. 2015, Brönnimann et al. 2019) has shown that flood events are mostly affected by precipitation on 3-4 days prior to the event. Although catchment size varies in our studies, Rx5day is expected to characterize heavy rainfall characteristics over a large range of catchments."*

Froidevaux, P., Schwanbeck, J., Weingartner, R., Chevalier, C., and Martius, O.: Flood triggering in Switzerland: the role of daily to monthly preceding precipitation, Hydrol. Earth Syst. Sci., 19, 3903–3924, https://doi.org/10.5194/hess-19-3903-2015, 2015.

Line 52: typo in controversy Thanks

Figure 2 and 3: "All series are smoothed with a 30-yr moving average". It looks like a backward smoothing, not centered (the data reach well into the 2000s). Maybe it should be explicated. Is the type of moving window considered when identifying the three multidecadal periods of flood variability analysed later on?

Thanks for this comment. No, it is centered, but we had calculated the filter to the end of the series (when only 15 data points were available). In the revised paper we set a minimum limit to 20 available data points. A special note is added to explain our averaging and filtering conditions:

*"Note that throughout the paper, a 30-yr moving average was used for visualisation, where at least 20 values must be available. For averaging regions we require that half of the regions have available data; only when averaging within regions we did not require a minimum as the chosen clusters were largely homogeneous such that the drop-out of a series will not have a large effect."*

Line 143-144: "selected from the 1x1° grid such as to best represent atmospheric processes relevant for the region)" is this based on some skill metric like correlation on some target? Some more words would help here, especially if in contrast with an alternative approach such as to spatially average the reanalysis data over several gridpoints.

The catchment sizes used in our study vary in size, so the averaging regions would be very different and shapes could be complex. Therefore, we opted for "point information", as for Rx5day. Furthermore, using 3 hourly atmospheric data on pressure levels from 80 members over a 210 year period requires a substantial amount of downloading, extracting and processing the calculation of entire fields. We did this only for selected grid points.

In the revised manuscript we added a parenthese note *"note that we preferred point data, as the Rx5day data also are point data"*

Line 171: check typo "for only for" Thanks

Line 214: PMIP maybe worth to be explicated We now write "Paleoclimate Modelling Intercomparison Project (PMIP)"

Line 232-233: Maybe this statement requires a bit more support. In my understanding, non-climatic anthropogenic influences on river runoff processes (e.g., river network changes, dams, etc.) may enhance/dampen multidecadal runoff variability or at least affect the autocorrelation of the discharge time series hence the detectability of multidecadal fluctuations above the red-noise background.

Thanks. This is correct. River works or new dams can induce step changes. One series (Rhine, Basel) was corrected for two such changes. Step changes do affect annual peak streamflow, an example is the Upper Rhone, which was excluded from our study (Hingray et al. 2010). The paper however also shows that large floods remained unaffected. We cite and discuss that paper briefly in the revised manuscript (added material in bold):

L 137: *"Other explanations include the role of power plants or other hydraulic constructions on the flood regime **(which is studied for the case of Porte-du-Scex, see Hingray et al. 2010)**."*

L. 269-272: *While numerous non-climatic factors (e.g., changes in the stream network and land use) contribute to long term trends **or may induce step changes (e.g., Hingray et al. 2010),** multidecadal variability is less influenced by such changes **(note that the Rhine series was corrected for two such changes)** and hence climatic conditions are analysed.*

Hingray, B., Schaefli, B., Mezghani, A. & Hamdi, Y. (2010) Signature-based model calibration for hydrological prediction in mesoscale Alpine catchments. *Hydrol. Sci. J.* **55**(6), 1002–1016.

Line 253-254: for me it was somehow difficult to check this statement by comparing the figures, especially given the premise provided in the preceding statements. I see that the documentary versus observational evidence is not central, but in its current form this aspect of the study appears to be missing some elaboration, either in the text or as additional analyses (for instance, I was just thinking that some bivariate wavelet analysis may work here).

Thanks, that is a relevant comment. Yes, the agreement is hard to spot for individual regions, which may be due to the fact that the Blöschl data are an interpolation. Interpreting them locally may be difficult. As mentioned in the text, we focus on some salient features: the flood-rich decades in the middle and late 19th century in Central Europe, in the early 20th century in Northwestern Europe, the Europe-wide flood-poor period after 1950, and the recent increase in flood intensity. Therefore, in the revised manuscript we show the comparison between peak streamflow series and the Blöschl series in a spatial aggregation:

Northwestern Europe (UK and Southern Norway) and Central Europe (all other regions). This is shown in a Supplementary figure (bottom part of Fig. S5). These salient features do appear in both series; at this aggregation level the comparison is much better.

*"This becomes clear when aggregating the series spatially into Northwestern Europe (UK and Southern Norway) and Central Europe (all other regions) and smoothing the Blöschl data for better comparability with the 30-yr smoothed streamflow (see Fig. S5)."*

Figure S4: the red line looks more like brown? On my screen it appears red.

Line 269-270: correlation of 0.21 appears rather low to me in terms of shared normalized variance, especially for smoothed/temporally aggregated time series, as I understand is the case here, which may contain a significant trend. Significance levels appear to be missing and should be provided, possibly accounting for autocorrelation of the series.

Yes, it is not very high (yes, the aggregation here was 4-yrs). We had used the 4-yr aggregation for reasons of consistency with the Blöschl comparison. However, we will replace this with correlation in the unsmoothed data (they are actually higher) add more details on the correlation test (it is a t-test, $p<0.05$, since always at least one record has no significant autocorrelation, the test is valid and autocorrelation does not need to be accounted for). In addition, the significant correlations are now highlighted in italics in Fig. 4. The relation between Rx5day, peak streamflow, and CONV5d is further evaluated using cross-wavelet analysis (new Fig. S6). This reveals an association mainly on the multidecadal scale:

*L. 345-8: "The association between the three series is further supported by cross-wavelet analyses (Fig. S6), which shows significant relations at time scales longer than ca. 30 years."*

Whether 0.21 is high or not is an interesting question. Recall that we compare the maximum annual peak streamflow with the maximum annual 5-day rainfall event. These will often not be the same events. For instance, in rivers with predominantly winter floods, the Rx5day will capture mostly summer events and the peak stream flow mostly winter events. In this case we to not expect a strong correlation. We add the following discussion:

*L. 319: "Note that a high correlation is not necessarily expected on a year-to-year scale as Rx5day events often do not occur together with annual peak streamflow. In winter flood regions, for instance, Rx5day occurs predominantly in summer, whereas peak streamflow occurs predominantly in winter, hence a winter series is correlated with a summer series."*

Figure 5a: how is the regional mean calculated? As I understand the calculation, as we move further back in the past, less time series contribute to the average, so this could lead to an inhomogeneity that can explain the discrepancy in the early period between time series. Possibly some illustration of standard error can reveal this uncertainty…

We used all available series in the submitted manuscript. In the revised manuscript, we plot (throughout the paper) only averages if at least 50% of regions have data. In addition, we added the standard errors to this plot. As commented above, we additionally use wavelet analysis to corroborate the association.

Figure 6b: can regions with non-significant regressions be indicated (for instance through shading)? Both map show a clear imprint of topography, which might be related as well to different variances in seasonal temperatures, significance would help to illustrate such effect for the T-FPI connection.

Yes, we added significance to the plot. In the summer case, almost the entire domain is significant. In the winter case, the cooling in the Alps is significant as well as the warming in the East.

Line 322: cyclonic weather type centered where? Over Switzerland (is now added)

Line 352-355: this may resemble a negative Eastern Atlantic rather than a negative NAO… Thanks, this is a valid point. We add this in the revised manuscript.

*"similar to a negative mode of the North Atlantic Oscillation or East Atlantic Pattern"*

Line 376: where is significance shown? Hatching is mentioned in the caption but I do not see it in the figure (rather I see red contours, that may encompass regions of significance?)

Yes, the orange line showed significance – we changed back to hatching (as stated in the caption).

Paragraph 3.4: This is another occasion when there is mostly a qualitative presentation of the comparison between different sources. Can this be improved?

Thanks. We added the ensemble spread for each line to Fig. 6 to show that the circulation effects are outside the ensemble spread. Since adding shadings to 8 lines on top of each other would produce a mess, we added the spread at the beginning and end of the period to the left and right side of the plot – the spread decreases approximately linearly.

Line 385: external forcing comes out a bit out of the blue here. Maybe some further elaboration would help.

The focus of the paper are not external forcings, but they are in the model boundary conditions, so we need to state that. We rephrased this.

*"(note that in addition to sea-surface temperature, they also encompass external forcings)"*

**Reply to reviewer 2**

The authors identify a "long-lasting conundrum" in the literature such as past flood-rich periods occurred mostly during cold conditions, while more floods are expected with the ongoing global warming. They develop an approach aiming to explore changes in the atmospheric conditions (dynamic versus thermodynamic processes) that could explain these distinct patterns in flood activity in Europe. The scientific question is highly relevant as e.g. flood projections still encompass large uncertainties, partly because how the climate change may regionally modify flood hazard is still unclear. The study is rather well designed and the paper well written. However, i) the "long-lasting conundrum" as presented here is not fully consistent, ii) the objectives and the way of proceeding for some treatment / analyses need clarifications and, most importantly, iii) the findings partly rely on visual analyses, limiting the robustness of the conclusions. These points are detailed in the comments below.

**Main comments**

1. The "long-lasting conundrum" (section Abstract and Introduction)

The authors introduce a "long-lasting conundrum" with the increased occurrence of floods during past cold periods and the expected increase of floods in the future with the climate change. This relies on the comparison between historical / paleodata and projections. However, the comparison is limited by differences in i) the time scales, ii) the studied catchments and iii) the return period considered. For instance, changes in flood activity from paleodata are mostly observed at longer timescales than those discussed here (centennial versus decadal). Paleodata also come from very small catchments (a few km²) compared to those studied here or those studied with historical data and projections (>1000 km²). Floods discussed in historical and paleodata are characterized by high return periods (>10-100 years), while the authors discussed here annual flood events. All these differences may result in a large range of flood-prone hydrometeorological processes and, thereby, in various responses in flood variability to the same climate change. This may easily explain this "long-lasting conundrum".

Thanks for the comments. This is an interesting point. We will reformulate the abstract, as our target are not these very long time scales. The author is right that time scales, catchments, and return periods are different. These three factors are discussed better in the revised manuscript at several instances. In the abstract we remove the sentence *"Here we reconcile the apparent contradiction"*. We also add a reference to a recent paper by Wilhelm et al. (2022) which reiterates these points and shows that several warming periods during the last 9000 years induced a decrease in the frequency of large floods (return period >10 years). Our return period is shorter (it is actually slightly longer due to the seasonal focus, which selects slightly stronger events). This is now discussed. For unstance, we add:

L. 87-89: *"Note that palaeoclimate studies are often based on events with a longer return period (e.g., 10 years or longer; Wilhelm et al., 2021)."*

Later in the discussion (L. 374-376): *"Although palaeoclimate studies often are based on small catchments, target a longer return period and a low-frequency variability scale that is longer than decades as in this study, it is nevertheless interesting to analyse the relation between temperature and floods on a multidecadal scale."*

And again in the Conclusions (L. 482-484). "*Note, however, that paleoclimatic studies often address longer time scales, smaller catchments, and longer return periods than are used in this study.*"

The water vapour effects is expected to always play a role – no matter whether a warming or cooling is forced or due to internal variability, and it is expected to not strongly dependent on the time scale. The circulation effect, however, may be different in the two cases, and it certainly depends on the time scale analysed.

In addition, the authors only use 1 reference about flood projections (l.55), while many other studies have been published and show large differences even in the sign of the change (some of these references are used l.421-422). Thereby, selecting another one may also show a decrease in flood activity under warming conditions. The authors may more convincingly use the recent findings of Blöschl et al. (2020, Nature) to introduce this "conundrum" – indeed, they showed that past flood-rich periods occurred under cooler conditions, while the most recent one occurred under warm conditions with a more homogenous dataset.

Thanks, it may be a better introduction to use the historical period rather than the Holocene period. We introduce this open question with Blöschl et al. by adding:

*L. 59-61: "Our paper addresses effects of atmospheric circulation changes and of climate warming on European floods on a multidecadal scale, following the work of Blöschl et al. (2020)."*

2. Treatment / analyses that need further explanations and/or quantifications

Discharge data (l.72-74) - The authors apply many treatments to the discharge data without explaining the rationale behind. This needs clarification. I am also wondering why the authors do not use the raw data instead of this kind of index of "flood intensity", especially when this provides similar results as stated.

In our study we aggregate many different series, and for this they need to have similar properties. The normalization does that (at least to some extent). This is explained in the revised paper.

*"Since in later steps, series will be aggregated, this transformation ensures that combined series have more similar properties."*

"Flood intensity" is simply our short name for "normalized peak streamflow". This can easily be changed.

Precipitation data (l.88-90) - The authors chose 2 precipitation indexes (Rx5d and Rx20d).

Again, there is neither a rationale nor reference to explain why the authors chose 5 and 20 days. The sizes of the studied catchment areas are very different and floods may be triggered by rainfall events of different durations. In addition, why a short and long duration? What do these two indexes represent here? This also needs clarification. At the end, only the Rx5d is used in the analyses.

Thanks for the question. The diagnostic for flood-propelling rainfall is Rx5day. We use Rx20day (more precisely, the seasonality of Rx5day) only for characterising the different streamflow series in the clustering process. Both is now better justified.

For typical river floods, shorter periods are relevant. We have checked this for one station (Basel) in a previous publication (Brönnimann et al., 2019) and found that the 2-3 days prior to the event are the most relevant (5 days prior to the event precipitation is already above the 75[th] percentile, but this is not extreme). There is a more systematic study on this by Froidevaux et al. (2015) concluding "that the consideration of a 3–4 days precipitation period should be sufficient to represent (understand, reconstruct, model, project) Swiss Alpine floods." Note that the size of catchments varies largely in our study; some are larger than

those studied in Froidevaux et al., some are of similar size. Eventually we aggregate series regionally (both precipitation index and streamflow). Hence, Rx5day should be a good choice. We added to the manuscript:

*"From the precipitation series we calculated Rx5day and Rx20day, i.e., the annual maxima of precipitation sum over periods of 5 and 20 days, respectively. The latter is used to characterize the seasonality of hydrological preconditions (e.g., soil saturation) in a catchment, as further discussed in the next section. The former is used as a diagnostic of flood-propelling events. Previous work (Froidevaux et al. 2015, Brönnimann et al. 2019) has shown that flood events are mostly affected by precipitation on 3-4 days prior to the event. Although catchment size varies in our studies, Rx5day is expected to characterize heavy rainfall characteristics over a large range of catchments."*

Froidevaux, P., Schwanbeck, J., Weingartner, R., Chevalier, C., and Martius, O.: Flood triggering in Switzerland: the role of daily to monthly preceding precipitation, Hydrol. Earth Syst. Sci., 19, 3903–3924, https://doi.org/10.5194/hess-19-3903-2015, 2015.

Flood seasonality (Fig. S1 and l.93 and following) - The authors perform a selection to get a set of cold- and warm-season floods, assuming they mirror distinct, regional hydrometeorological processes. The considered seasons are here long of 6 months. Why 6 rather than 1 or 3 months needs to be explained.

From an atmospheric point of view, heavy rainfall is associated with specific weather patterns such as elongated troughs or cut-off lows (see Stucki et al. 2012). Winter events tend to be related to different pattern (e.g., zonal flow) than summer events. Moreover, the role of convection is stronger in summer. It therefore makes sense to discriminate a cold and a warm season, but a finer partitioning would probably not result in more different weather regimes. Conversely, it would strongly decrease the sample size. In the revised manuscript we added the text:

L. 149. *"Seasonality is an important factor to consider as it is characteristic for a given region. Furthermore, the relevance of atmospheric process changes in the course of the year. Winter events tend to be related to different circulation patterns (e.g., zonal flow) than summer events (Stucki et al. 2021). Moreover, the role of convection is stronger in summer. In the following we will therefore perform all analyses for annual data as well as for annual series restricted to flood seasons, defined as May ro October (for clusters Upper Rhine and Danube) and November to April (all other clusters). This partitioning captures the seasonal flood characteristics as well as the seasonal differences in atmospheric processes and it still ensures an adequate sample size."*

The authors also discard 5 flood series because their triggers may include e.g. snow processes (l116-117). However, much more series do not show a good correspondence between the highest values in the precipitation indexes and the highest occurrence of the annual peak streamflow (Fig. S1), suggesting that almost half of the series represent mostly floods triggered by a mix of processes in which precipitation is not a dominant driver. Or that the chosen precipitation indexes are not the most relevant. The selection process of the series is thereby questionable.

The strongest precipitation event in a year does not necessarily cause the highest streamflow. As explained in the paper, other factors contribute. We would like to avoid selecting flood events for which the atmospheric disposition was not relevant; this would dilute our sample. We also do not want to be select only a handful out of 47 series. We have good reasons for removing the five series. The corresponding paragraph, which was already in the original manuscript, is rephrased:

*"… Other explanations include the role of power plants or other hydraulic constructions on the flood regime (which is studied for the case of Porte-du-Scex, see Hingray et al. 2010). In any case, since the focus of this study is on atmospheric processes, these rivers might confuse our results and hence we removed five series from the three clusters (Inn at Martinsbruck, Rhône at Porte-Du-Scex, and Rhine at Domatems, Neuhausen, and Rekingen)."*

It is correct that the remaining series are not purely atmospherically driven, and a comments on that is added in the context of correlations between Rx5d and peak streamflow.

L. 323: *"Nevertheless, years with high peak stream flow coincide with years with high maximum 5-day precipitation, although the association is not very strong and one needs to keep in mind that flood intensity is not purely atmospherically driven."*

Correlation test (l.248 and following, Fig. 4) – A correlation test (but which one is not indicated) has been performed between peak streamflow and Rx5day (why not also with Rx20d?). Among the results shown (8 rivers among 43?), the values are rather low for most of them (< 0.35). First, results for the other results should also be shown in e.g. a table in Supplementary Material so that the reader can have an overview of its relevance.

Thanks. The correlation test will be better explained. It is a t-test and was performed at the level of the regions for Rx5d (as mentioned above, Rx20day was only used for discriminating series), not at the level of the individual rivers. But in the revised manuscript we now add a column to Table S1 with this information on the level of individual series. In the submitted manuscript, we performed the correlation for 4-yr average, to be consistent with Blöschl. However, this introduces a new time scale, which may be confusing. Therefore, in the revised manuscript, we use only the unfiltered data for the correlation (and then use cross-wavelets to address the relation between two series as a function of time scales).

Correlations (on the unfiltered data) are in the range of 0.35-0.4. Whether this is low or high is another question. As mentioned above, the strongest precipitation event in a year does not necessarily cause the highest streamflow. Some series will have their strongest Rx5day event always in summer and the highest peak streamflow always in winter. So, we essentially correlate a summer series with a winter series. Do we expect a higher correlation than 0.3 between the two? This is another reason for the seasonal stratification. We now give more explanation of this in the revised manuscript.

The rephrased paragraph (L. 16) reads: *"First, we analysed the relation between flood intensity and precipitation. In most regions, flood intensities are statistically related to Rx5day. Correlations (Figa. 4 and S5, calculated from annual data) vary greatly (between 0.05 and 0.7), but are significant (t-test, p<0.05) for six regions. Note that a high correlation is not necessarily expected on a year-to-year scale as Rx5day events often do not occur together with annual peak streamflow. In winter flood regions, for instance, Rx5day occurs predominantly in summer, whereas peak streamflow occurs predominantly in winter, hence a winter series is correlated with a summer series."*

Second, the general low values suggest that precipitations explain only a small part of the variability, limiting the relevance of the following analyses to explain changes in flood variability. This point is not discussed. About the correlation between CONV5d and peak streamflow, it is only assessed visually, while it is a key link for the following analyses.

As mentioned above, the low correlation on unfiltered (or 4-yr filtered) data does not mean that precipitation is unimportant. Annual maxima of Rx5d and annual maxima of peak streamflow often capture different events, so their time series may not be strongly correlated (we do expect some correlation). We ask whether there is a multidecadal variation that is common to both, which would then lead to stronger correlations on a low-frequency scale. In

addition to the visual judgement, we now also perform cross-wavelet analyses (new Fig. S6) and add the sentence:

L. 346-348: *"The association between the three series is further supported by cross-wavelet analyses (Fig. S6), which shows significant relations at time scales longer than ca. 30 years."*

Similarly, the respective contributions of the circulation change, water vapour change and interactions on changes in the annual peak streamflow is also based on visual "correlation" (l. 294 and following; Fig. 5). Therefore, the findings mainly rely on visual comparisons, strongly limiting their robustness. Instead, we expect that a correlation test as well as a significance test to be applied systematically to each correlation discussed and supporting the findings.

All correlations are tested and significance is highlighted by italic font. For the case of Fig. 5. We perform cross-wavelet analyses of the unfiltered data see comment above). These analyses confirm significant relations on multidecadal scales between Rx5day and streamflow, between CONV5d and Rx5d, and between CONV5d and streamflow. So the conclusions is not just based on visual analysis.

3. The relative contribution of atmospheric processes to changes in peak streamflow

The authors stated that periods with higher flood intensity prior to 1950 are mainly due to circulation changes, while the period with higher flood intensity after 1950 is more related to changes in water vapour changes. However, looking at Fig. 5b, the contribution of circulation changes is also increasing after 1950, better mirroring the increase of CONV5d than water vapour changes. Therefore, a quantification of the respective contributions of atmospheric processes to changes in CONV5d is really needed to objectively assess them.

In the revised manuscript we quantify this by addressing the trend since 1963 (when streamflow reached a minimum) in CONV5d and in its contributions. This is based on unfiltered data (as all analyses - filtering is only used for plotting). This shows that CONV5d increases, the contribution of circulation changes has no trend, wheras the contribution form water vapour changes increseases even more strongly than CONV5d and is highly significant. This is added to the manuscript:

L. 357: *"An analysis of linear trends in the unsmoothed series since 1963, the minimum in flood intensity, reveals an increase in CONV5d ($4.04 \times 10^{-7}$ kg m$^{-2}$ s$^{-1}$ yr$^{-1}$, which is not statistically significant), no trend in the contribution of atmospheric circulation changes, but a highly significant increase in the contribution of water vapour changes ($6.13 \times 10^{-7}$ kg m$^{-2}$ s$^{-1}$ yr$^{-1}$)."*

In addition, the authors state that this explains why flood-rich periods have been mostly observed during cold periods in the paleodata. However, large changes in temperature have also been reconstructed over the last millennia. So, we may also expect that changes in water vapour played a role further back in time?

Exactly! Any change in temperature, irrespective of its underlying cause, would be expected to cause higher water vapour concentrations and therefore higher CONV5d. This is what we show in Fig. 5b. However, if that temperature change is caused by a change in circulation, then this circulation change (in our study regions and seasons) operates in the opposite direction. Because circulation dominates at the multidecadal times scale we focus on, we do not see the temperature effect in the past (only our decomposition uncovers this). In contrast, the water vapour contribution becomes obvious in recent years because the warming was not (or not strongly) driven by circulation changes and therefore circulation does not counteract the water vapour signal in this case. This leads back to the initial comment of this reviewer: It

is indeed an open question whether the same holds for centennial variations. It may turn out that on these scales, water vapour changes always dominate.

In the revised manuscript (L. 394) we add "*Warming phases (also in the past) rather directly lead to an increase in CONV5d, but warming may be driven by atmospheric circulation changes that decrease CONV5d, or it may be driven by other forcings in which case atmospheric circulation does not counteract the increase in CONV5d.*"

**Minor comments**

L.117. The authors removed five series. They should name the series they removed. (they are now mentioned)

173. Typo: "in order to" Thanks

l.180. Why 5 days? What this duration correspond to? This window length and weighting was taken from a previous study (Brönnimann et al., 2019) and was based on analyses of daily discharge, precipitation, and water flux convergence on the preceding days. For the current study we had also tested using varying window lengths l depending on catchment size, using an equation l = sqrt(A)/47.3 + 2. Results were very similar. We also tested using different windows for water flux convergence. As the methods are already very complex, we chose not to elaborate on all the myriads of tests we performed. In the revised manuscript we add the sentence:

L. 213-215: *"This window length and weighting was taken from a previous study (Brönnimann et al., 2019) and was based on analyses of daily discharge, precipitation, and water flux convergence on the preceding days."*

l.221. The authors may refer to Fig. S1 instead of S2? Yes, correct, thanks.

l.227. Typo "such less pronounced peaks" Thanks

230. "1919-.. exhibit low values)". This is unclear. Changed to: local maximum at a time when warm-season series exhibit low values

248. "at 4-yr aggregation" again, why 4? This is the resolution (voxel size) in Blöschl et al. (2020). This is now mentioned in the revised manuscript (L. 287/8).

Fig. 4 and S4 are very similar. So, Fig S4 could replace Fig. 4. The agreement demonstrates the robustness of the results. We prefer to show the seasonal analysis in the paper (Fig. 4) and the annual analysis in the Supplement for the reasons stated above (we are then more sure to have captured the relevant atmospheric processes, even though the result is the same). We think that it is important not to mix winter-dominated flood series with summer-dominated Rx5d series.

Fig. 7. Why the 80 members are not shown here? What is the curve, the mean of the 80 members?

Yes, it is the mean. This is changed in the caption. As the figure shows 32 time series, it is challenging to add the ensemble standard deviation to each one. In the revised manuscript we add bars (centered around zero) at the beginning and at the end of the series with their length corresponding to 2 ensemble standard deviations (+/-1). As the standard deviation decreases approximately linearly from the beginning to the end, it is sufficient to show it for the first and last year. This is added in the revised manuscript. At the same time, the figure confirms that the multidecadal variations of the circulation contribution varies by more than 2 standard deviations.

Fig. S1. The way the seasonality is identified is sometimes misleading since there is not always a correspondence between the six months of high precipitation and high discharge. Thereby, for some series, months with the most frequent annual peak flow are not considered.

There are two streamflow series where the month with the maximum peak streamflow is not in the selected flood window. The clustering put these two series in the same clusters as other series with a different seasonal cycle. Note that we perform all analyses also for annual series and show the results in the supplement.

Fig. S3. Why these series are excluded from the analyses? This was explained in the paragraph l. 105-127 and Figure 2: Five of them concern high-altitude catchments that are either affected by snow melt or by hydraulic installations, but in any case do not show the same flood regime as on the same further downstream. The sixth turned out to be a one-series cluster and is reportedly affected by snow melt and rain-on-snow events. We nevertheless did not want to hide the results from these, and so put in a supplementary figure. In the revised manuscript we add to the Caption of this supplementary figure: (see Section 2.2 and Fig. 2). Furthermore, we add another reference supporting the effect of hydraulic installations in one case (Hingray et al.).

Fig. 1a. Why the numbering of the discharge series starts by #2 (instead of #1)? SI Table 1: why the cluster column is empty? If so, it can be removed.

Thanks should be #1, then #3. The numbers of the cluster are added.

Thanks for this very careful review and for the thoughts about the difference of our setting and that of palaeostudies.

---

## Author Response (AR2)

Dear Editor

Thanks for sending the review. As requested by the reviewer, we checked the effect of smoothing in the presence of missing values and did not find an impact on the results.

Thanks for handling our manuscript.

Best regards

Stefan Brönnimann